# Predicting Emotion and Engagement of Workers in Order Picking Based on Behavior and Pulse Waves Acquired by Wearable Devices

**DOI:** 10.3390/s19010165

**Published:** 2019-01-04

**Authors:** Yusuke Kajiwara, Toshihiko Shimauchi, Haruhiko Kimura

**Affiliations:** 1Department of Production Systems Engineering and Sciences, Komatsu University, Shichomachi Nu1-3, Komatsu, Ishikawa 923-8511, Japan; haruhiko.kimura@komatsu-u.ac.jp; 2Department of Creative Community, Komatsu College, Shichomachi Nu1-3, Komatsu, Ishikawa 923-8511, Japan; shimauchi@komatsu-c.ac.jp

**Keywords:** order picking, emotion, engagement, the wearable sensor, deep neural network, flow experience

## Abstract

Many logistics companies adopt a manual order picking system. In related research, the effect of emotion and engagement on work efficiency and human errors was verified. However, related research has not established a method to predict emotion and engagement during work with high exercise intensity. Therefore, important variables for predicting the emotion and engagement during work with high exercise intensity are not clear. In this study, to clarify the mechanism of occurrence of emotion and engagement during order picking. Then, we clarify the explanatory variables which are important in predicting the emotion and engagement during work with high exercise intensity. We conducted verification experiments. We compared the accuracy of estimating human emotion and engagement by inputting pulse wave, eye movements, and movements to deep neural networks. We showed that emotion and engagement during order picking can be predicted from the behavior of the worker with an accuracy of error rate of 0.12 or less. Moreover, we have constructed a psychological model based on the questionnaire results and show that the work efficiency of workers is improved by giving them clear targets.

## 1. Introduction

Many logistics companies adopt a manual order picking system. Order picking is a task of picking items on the shelves according to the order form and delivering them [1]. To reduce costs [2], related research has been conducted to reduce human errors and improve the work efficiency of workers. This related research has revealed the effects of stress, motivation, and work satisfaction on order picking. The worker often experiences a positive flow when the difficulty of work and the ability of the worker are balanced and he/she receives a clear target and feedback. The worker in a flow experience is engaged in the task and evokes a pleasant emotion. There are two kinds of affect: emotion and mood. Emotions are defined as affective states that change during a short time. Mood is defined as a relatively long-lasting affective state, which is different from emotion, as mood is less specific and less tense [3]. Engagement is defined as when “all the attention of workers is directed to completing a task, and workers feel the transformation of time sense” [4]. Pleasant emotions increase the concentration of a worker and encourage positive behavior [5]. Furthermore, pleasant emotions have an effect to increase the immune strength of the whole body, and they contribute to the maintenance of the physical health of the worker [6]. Engagement improves the performance of the worker [7] and gives a sense of fulfillment [8,9]. Pleasant emotions and engagement contribute to a higher job retention rate [10,11]. By managing a worker’s psychological state, the organization can maintain the motivation of workers and improve the productivity of order picking. However, it is difficult to measure the psychological state after each order picking using a questionnaire because it interrupts the work and reduces the productivity of the work. In addition, often when we conduct a questionnaire frequently, the worker will embrace unpleasant emotions and distracts attention from the work. Therefore, it is desirable to develop a method to predict the psychological state of the workers for each task with less burden on the worker.

In this research, we propose a system that utilizes wearable sensors and deep neural networks to predict the degree of worker’s emotion and engagement. First, behaviors and pulse waves are acquired with wearable sensors. Second, time series features are calculated from the behaviors and pulse waves. Third, we predict the emotion and engagement by inputting the time series feature to deep neural networks. In this research, emotions are represented by Russell’s circular model of affect [12]. Finally, we maximize the work efficiency and minimize the human error of workers by improving work environment based on the prediction results and psychology model. The contributions of this research are as follows:(1)Emotion and engagement of workers were predicted from behavior and pulse waves acquired with wearable devices and the accuracy of the prediction was clarified. Furthermore, we identified important variables to predict emotion and engagement.(2)We construct a psychology model of worker and clarified effects of emotion and engagement.

Section 2 describes the definition of flow experience, the method of measuring the psychological effects of flow with references to related research. Section 3 describes the method to predict the emotion and engagement using wearable sensors and deep neural networks. In Section 4, we show the prediction accuracy and clarify the explanatory variables which are important in predicting the emotion and engagement during work with high exercise intensity. In addition, we show the validity of psychology model of the worker from questionnaire results. Section 5 draws conclusions and outlines the future directions of our work.

## 2. Related Research

### 2.1. Definition of Flow Experience

A worker measures flow experience by a flow state scale [13] (Table 1). The scale consists of a balance of task difficulty and worker skill, a fusion of behavior and consciousness, control behavior, instruction of clear target and feedback, concentration, a transformation of time sense, evocation of emotion.

Each item of the flow state scale is evaluated on the Likert scale [14] with 5-point scales or 7-point scales. The degree of flow experience is evaluated on the average value of items of the flow state scale. The evocation of emotion is measured by the circumplex model of affect [12] and the general affect scale [15].

Flow experience has been shown to be effective in various fields. Flow experience occurs mainly when the task difficulty and the worker’s skill are balanced [16]. When this balance is achieved, the worker is fused with behavior and consciousness, which reduces the cognitive load and improves job performance [17]. The worker who behaves autonomously with a clear goal has high job satisfaction [18,19]. The worker with flow experience evokes pleasant emotions and displays increased subjective well-being [20,21].

### 2.2. Measurement of Emotion

Emotion is measured from behavior and vital signs using wearable devices. For the indicators of behaviors, the pitch of the voice [22] and facial expressions [23] have been used. For the indicators of the vital signs, pulse rate variability, respiratory variability, and the electrodermal activity have been employed [24,25,26]. These indices directly or indirectly express the degree of activity of autonomic nerves. Organisms have the property of controlling the autonomic nervous system when an external/internal stimulus is applied and keeping physiological parameters (heart rate, blood glucose level, etc.) within an allowable range. This function is called homeostasis [27]. Internal stimuli include affects such as anxiety and anger. Therefore, affects are predicted by measuring the activity of the autonomic nervous system controlled by homeostasis. However, the physiological parameters kept by homeostasis change depending on the type and condition of stimulation [28]. Therefore, when the same stressor is given under different condition, the response of the vital sign often differs. In a related research, a video was shown to the subject to evoke a specific emotion [29] and predicted the emotion from the change of the vital signs and behavior on real-time basis using wearable devices [30]. However, these researches were directed to emotions of resting subjects. Since the vital signs and behavior is an indirect indicator of emotion, it is not always possible to estimate the emotion during order picking by the indicator of the vital signs and behavior at rest. A related study predicts pleasant and unpleasant moods in everyday life with an accuracy of 60% by acquiring gait and pulse rate variability using a wristwatch pulse wave meter and smartphone [31]. However, this related study predicts the mood of the day, Hence, it is difficult to predict the emotions that change in a short time like during order picking.

Since the worker performs a task alone and is appraised by the outcome, there are few workers that talk to others worker and change facial expressions during order picking. In addition, the worker works in order picking while moving the whole body. Therefore, it is meaningful to explore new variables for effectively predicting emotions in order picking.

### 2.3. Measurement of Engagement

The effects of engagement on the worker are less cognitive load and full attention to the task. The cognitive load was defined as the mental resources a person possesses for solving problems or completing tasks at a given time [32]. The cognitive load and attention are measured from behavior and vital signs using wearable devices such as inertial sensors [33,34], eye tracker [35,36], electroencephalographs [37], pulse wave meters [38]. A related research loaded working memory [39] of subjects on cognitive tasks and attention task such as semantic fluency task, elementary cognitive tests, and visual search task, and predicted the cognitive load from the change of the vital signs and behavior on a real-time basis using wearable devices [40]. The related research estimated the user’s cognitive load and attention during rest such as personal computer work [40] and driving a car [41]. In addition, the related research predicted the cognitive load of pedestrians using inertial sensors attached to the waist. The related research predicted engagement from behavior and vital signs when a worker is at rest. There are studies that predict cognitive load during action, but its behavior is simple. The worker performs complicated behavior. Therefore, it is meaningful to explore new variables for effectively predicting engagement during order picking.

## 3. Predicting Emotion and Engagement Using Deep Neural Networks

Figure 1 shows the prediction of emotion and engagement using deep neural networks. In this research, we calculate features from behaviors and pulse waves acquired from wearable devices and input them into deep neural networks, thereby predicting the emotion and engagement. The emotion is represented by a circumplex model of affect as illustrated in Figure 2. In the circumplex model of affect, emotions are represented by mutually orthogonal pleasant-unpleasant axis and arousal-sleepy axis. Workers can easily describe their current emotions by simply marking the corresponding square in the model, which includes 7 × 7 squares.

The model drastically shortens the time required for a subject to answer compared with PANAS [15]. PANAS is a questionnaire asking about five positive emotions (relaxed, happy, concentrated, interested, and active) and five negative emotions (tired, stressed, sleepy, angry, and depressed). Therefore, the model is appropriate for situations where we often ask workers about their current emotions. The prediction values of emotion and engagement are visualized and used as an indicator of business improvement. Business improvement is conducted based on a psychology model.

### 3.1. Behavior and Pulse Wave Feature Calculation

To predict the emotion and engagement, we acquired the behavior and pulse wave of the worker from the beginning to the end of assigned task by using wearable devices. In this research, we use a pulse wave sensor, the motion detectors, and an eye tracker to acquire the time series data listed in Table 2. When the workers evoked an emotion, they activate the autonomic nerve and change the pulse wave [24]. When the workers’ pay full attention to a task, they change the movement of the waist while walking [33], pulse rate variability [38], pupil diameter [35], head movement [35], and eye-gaze [36]. The eye tracker used was the Tobii Pro Glasses 2 made by Tobii (Tokyo, Japan). The tracker is equipped with an acceleration sensor, an angular velocity sensor, two cameras for recording eye movements and viewpoint. The sampling rate of the cameras was 40 Hz. The acceleration and angular velocity of XYZ coordinates were acquired. The sampling rate of the acceleration sensor and angular velocity sensor was 100 Hz. The pulse wave sensor is a vital meter made by TAOS (Kanagawa, Japan). The sensor is equipped with a photoelectric pulse wave meter, an acceleration sensor, and an angular velocity sensor. The pulse wave meter acquires a pulse wave. The sampling rate of the pulse wave were 1000 Hz. The RR interval is calculated from the pulse wave. The frequency component of the RR interval is defined as very low frequency (VLF) at 0.04 Hz or less, low frequency (LF) at 0.04 Hz to 0.15 Hz, and high frequency (HF) at 0.15 Hz to 0.4 Hz. The combined power spectrum density of VLF, LF, and HF is defined as total power (TP). The acceleration and angular velocity sensors were put in the breast pocket. The sampling rate of the acceleration and angular velocity were 1000 Hz. Entropy and Lyapunov exponent are calculated from the acceleration and angular velocity. To acquire the time series data of VLF, LF, HF, TF, Lyapunov exponent, and entropy, time window was prepared and slid the time window at 1 s intervals. The time window size was 30 s. Therefore, the sampling rate of the VLF, LF, HF, TF, Lyapunov exponent, and entropy was 1 Hz. The motion detector is Neuron made by NOITOM (Beijing, China). The sampling rate of the motion detector was 50 Hz. The pulse wave sensor, eye tracker, and motion detector have sufficient sampling rate to calculate time series features.

According to Table 2, the task complexity and time series features are calculated from items and time series data of the behaviors and pulse waves. The mathematical symbols corresponding to the time series features calculated in this research are shown in Table 3. Type described in *u*^(*type*)^ means type of time series data acquired by eye tracker such as eye movement in XYZ coordinate system(3D), point of view(2D), and pupil diameter (PD). Type described in *p*^(*type*)^ means type of time series data acquired by pulse wave sensor such as VLF, LF, and HF. Part in described in *a*^(*part*)^*_axis_*, *g*^(*part*)^*_axis_*, and *r*^(*part*)^*_axis_* means the part of the body to which the sensor is attached. The time series features are average, standard deviation, minimum value, maximum value, the power spectrum density, peak-valley value, and entropy. The power spectrum density, which represents the periodic feature of the signal, is calculated by Fourier transform. In this research, the frequency resolution was set to 1/10th of the Nyquist frequency. The peak-valley value, which represents the kurtosis of the signal, is calculated by the following procedure: First, the extremum of the signal is acquired. Second, the acquired extremum is normalized to the maximum value. Third, we apply a threshold to the extreme and gain a significant change in the signal. When *extremum* ≦ 0.5 + *threshold* or *extremum* ≧ 0.5 − *threshold*, the extremum is adopted. Finally, the average value, standard deviation, detection time, minimum value, and maximum value are calculated from the adopted extremum. In this paper, time series features were described with reference to Table 2 and Table 3. For example, *σ*(*v*(*e*(*a*^S3^),0.25)) means the standard deviation of the peak valley value of threshold = 0.25 of the entropy of the cervical spine acceleration.

The power spectrum density of VLF represents breathing variability. The power spectrum density of LF represents the activity of the sympathetic and parasympathetic nervous systems. The power spectrum density of HF represents the activity of the parasympathetic nervous system. An excited worker activates the sympathetic nervous system and increases the power spectrum density of LF. A relaxed worker activates the parasympathetic nervous system and increases the power spectrum density of HF. The power spectral density of LF and HF represent the relaxed interval of the worker. The Lyapunov exponent and entropy, which represent the stability of order picking, are calculated from the acceleration sensor attached to the pulse wave sensor. Pulse waves change with behaviors and emotions evoked. Lyapunov exponent and entropy change with behaviors. Therefore, the power spectrum density of the pulse wave sensor shows that the emotions and behaviors change at a regular interval. In addition, the peak-valley value of the pulse wave sensor represents a significant change in behaviors and emotions.

The motion detectors acquire the rotation angle of the body joints. The rotation angle of joints changes with behaviors. The power spectrum value of the rotation angle of joints represents the rhythm of the work. In addition, the peak-valley value of the rotation angle of joints represents a significant change in behaviors.

The eye tracker acquires the eye-gaze, the pupil diameter, the number of saccades, and the number of blinks. The movements of the eye-gaze and the head change with order picking. The power spectral densities of eye-gaze and head represent the rhythm of work. The peak-valley values of eye-gaze and the head represent a significant change in behaviors. The pupil diameter represents a change of emotion. The power spectrum density of the pupil diameter shows that the emotion changes at a regular interval. The peak-valley value of pupil diameter represents a significant change in emotion. The number of blinks represents the degree of arousal.

The task complexity is represented by items and degree of overlap of items listed in the order sheets. When an area *j* is listed in order sheets, a dummy variable *l_j_*(*j* = 1, 2, …, 6) is set to 1, otherwise, it is set to 0. Second, to calculate the degree of overlap, when an area *j* is listed in an order sheet *l*, a dummy variable *l_j_*^(l)^ is set to 1, otherwise, it is set to 0. The degree of overlap is calculated as the average *m*(*l*) of *l_j_*^(l)^ and the standard deviation *σ*^(l)^ of *l_j_*^(l)^. In addition, we acquire working time *t*. In this research, there are six areas in the experiment environment. Therefore, the task complexity is represented by nine dimensions which combine *l_j_*, *m*(*l*), *σ*^(l)^, and *t*. The task complexity is added to the time series features of the pulse wave sensor, the motion detectors, and the eye tracker.

### 3.2. Variable Selection by a Hypothesis Test

From Table 2 and Table 3, we select variables effective in predicting the emotion and engagement by the Brunner Munzel test. First, an emotion and engagement are divided into two groups. The horizontal axis is divided into pleasant (PL) and unpleasant (UPL). We defined as UPL when the value in the pleasant-unpleasant axis was less than 0, otherwise, it was defined as PL. The vertical axis is divided into arousal (AR) and sleepiness (SE). We defined as SE when the value in arousal-sleepy axis was less than 0, otherwise, it was defined as AR. Engagement is divided into boredom (BR) and concentration (CR). The engagement was measured with a value of 1 to 5. We defined as BR when the questionnaire of engagement was less than 3, otherwise, it was defined as CR. We make a null hypothesis that there is no difference in a feature element of both groups of an emotion and engagement and conduct the Brunner Munzel test. The rejection region is 0.01. When the null hypothesis is rejected, a feature element has a significant difference between the two groups. Therefore, the feature element is selected as a variable effective in predicting emotion and engagement. Explanatory variables are consisting of the feature elements selected by Brunner Munzel test.

### 3.3. Prediction of Emotion and Engagement by Deep Neural Networks

In this research, we construct a regression model using deep neural networks. Deep neural networks using auto-encoder [42] learn the explanatory variables when emotion and engagement occurs. Deep learning has a high feature extraction ability, and excellent prediction accuracy are shown in papers of various fields. Therefore, we use deep neural networks to predict pleasant emotion, arousal, and engagement. Deep neural networks consist of an input layer, hidden layers, and an output layer. The explanatory variables are inputted to the input layer. First, deep neural networks adjust the weight of each unit in layers using auto-encoder. The auto-encoder adjusts the weight to output the same signal as the input signal. Therefore, in the deep neural network of the auto encoder type, the features of the input layer are aggregated and input to the output layer. Deep neural networks adjust the weight of each unit in the output layer by fine-tuning.

Deep neural networks calculate the effect size of each explanatory variables from the weight of each unit in layers. A variable with a high effect size is an important variable to predict emotion and engagements. The followings are the parameters for this study: Hidden layer size was 3. Unit size was 100. Activation function was Rectifier. Dropout ratio was 0.5.

## 4. Experimental Results

First, to show that emotion and engagement contribute to the productivity of the order picking, we construct the psychological model from the questionnaire results and clarify the effect of emotion and engagement on work efficiency and human error. Second, we show the accuracy of prediction of pleasant, arousal, and engagement by deep neural networks. Then, it shows that pleasant, arousal, and engagement can be predicted from behaviors and pulse waves acquired by wearable devices. Furthermore, we identify important variables to predict pleasant, arousal, and engagement.

### 4.1. Experimental Procedure

We conducted the experiment in an environment illustrated in Figure 3a. The subjects, 15 males and two females, were in good health condition. Subjects did not have order picking experience before the experiment. They were paid by the hour. The subjects picked items according to lists. The items were installed in the item box. The items to be picked were put in a delivery box on a hand truck. The subject visited the work area only once and picked the items. We instructed the subjects to finish order picking within one minute. After each picking session, the subjects answered pleasant, arousal, engagement, and subjective task difficulty experienced during the session. The subjects plotted the emotions into the 7 × 7 squares according to the circumplex model of affect in Figure 2. The subjects answered the level of engagement and subjective task difficulty in 5-point scales.

The delivery boxes and the lists are prepared for company 1–3. We prepared each product list for each delivery BOX. The number of areas was set to six, from Area A to F. Each item box contained four items to be picked. The size of the item box and the delivery box was 250 mm × 160 mm × 150 mm. To adjust the complexity of order picking, we created a simple list and a complex list according to the following rules. The type of items and the number of items is determined by a uniform random number. The maximum number of items is 3. The subjects should pick all the items on the list by visiting three areas. A simple item list is set so that the items and areas do not overlap between lists. A complex list is set so that the items and area overlap between the lists.

The subjects ran 32 order picking sessions. The subjects rested for 10 min after the 16th session to remove the effects of physical fatigue. Sessions from 1st to 16th are defined as the first period, and from 17th to 32nd as the second period. To verify the effect of the target on flow experience, we gave a clear target before the second period began. The subjects are told that “the wages are paid in full for the first period regardless of the outcome of work. However, for the second period, the wages are adjusted according to the outcome of work”. The merit-based pay system is widely used in various fields and clarifies the intention of the management to achieve a target. To clarify the effects of the target, we objectively evaluated the outcome by evaluating human error and work efficiency. The human error refers to the number of items that are picked incorrectly. Work efficiency is calculated by the Equation (1):*w* = (1 − *t*/60),(1)
where *t* is the working time. Since the worker performed order picking within 60 s, we divided *t* by 60. *t* and human error were recorded after each order picking. However, in fact, the wages are paid in full for the first and the second periods regardless of the outcome of work. We explained the fact to the subjects and got the consent of subjects.

To verify the effect of feedback on flow experience, the subjects were randomly given feedback in the first or the second period. All the feedback was positive. Nine subjects received feedback in the first period and the remaining eight subjects received feedback in the second period.

To verify the effect of flow experience, the subjects responded to the items of the flow state scale [13] in Table 1 at the ends of the first period and the second periods based on their task performance. The subjects wore an eye tracker, a pulse wave sensor, and motion detectors as illustrated in Figure 3b. Table 3 shows that the time series features are calculated from the acquired signals by the wearable devices.

### 4.2. Validation of the Psychological Model

Table 4 shows the sample sizes of each square in the circumplex model of affect. The largest square is the first quadrant, followed by the second, the fourth and the third quadrants. From Table 4, there are many in the order of the first quadrant, the second quadrant, the fourth quadrant, and the third quadrant of circumplex model of affect.

We analyzed the effect of subjective task difficulty on engagement. Table 4 shows the sample sizes of the questionnaire result of subjective task difficulty and engagement. From Figure 4, the engagement level of 3 and 4 occurred most frequently when subjective task difficulty was 0, while their occurrence decreased when subjective task difficulty is less than 0 or greater than 0. The engagement level of 1, 2, and 5 were uniformly distributed regardless of subjective task difficulty.

Subjective task difficulty and engagement were uncorrelated. From the above, adjusting the subjective task difficulty is effective to make the worker reach engagement level of 3 and 4. From these results, it is suggested that engagement level of 3 and 4 of the workers can be controlled by adjusting subjective task difficulty. However, the engagement was 1, 2, and 5 of the workers cannot be controlled by adjusting subjective task difficulty.

The psychological model was constructed from the results of the hypothesis test and factor analysis and it has verified the goodness of fit of the model by covariance structure analysis. The results of the hypothesis test and factor analysis are shown in Appendix A.

Predicted emotion and engagement contribute to the maximization of work efficiency and minimization of human error. The sample size was 544 (the number of subjects × the number of session = 17 × 32). The goodness of fit of the model was evaluated by the chi-square test. As a result, the *p*-value was 0.61, and the null hypothesis was adopted. The comparative fit index (CFI) was 1.00, and the Tucker-Lewis index (TLI) was 1.00. RMSEA was 0.00. We make a null hypothesis that RMSEA is less than 0.05 and tested. As a result, since the *p*-value was 1.00, the null hypothesis was adopted. The results proved the validity of the model in Figure 5.

Since the *p*-value of the effect of the complexity of the task on engagement and flow experience was 0.05 or more, the effects were not statistically significant. Since the *p*-value of the effect of feedback on flow experience was 0.05 or more, the effect was not statistically significant. The results were equal to engagement decreased as subjective task difficulty is less than 0 or greater than 0. Therefore, the results show that flow experience is adjusted by the subjective task difficulty. Pleasant emotions gave vitality to workers and reduced human error. Target improved work efficiency. Flow experience mainly affected arousal and engagement. On the other hand, pleasant emotion was affected by flow experience and complexity of the task. The effect of complexity of the task on pleasant emotion suggests that negative emotions were evoked when complicated work was given while wage is reduced by the outcome. It suggests that pleasant emotion is changed by the cognitive process of the worker.

The related research [43] has shown that flow experience mainly occurs when the task difficulty and worker’s skill are balanced. On the other hand, the effect of flow experience on work efficiency and human error in order picking is not shown. The effect of feedback depends on the way of feedback. The effect of feedback depends on how the feedback is given to a worker. Human error is affected by the complexity of the task and pleasant emotions. Flow experience does not directly affect human error. However, Figure 5 showed that pleasant emotion evoked by flow experience affects human error. Therefore, organizations and supervisors can reduce human error by managing the flow experience of the worker. Since the complexity of the task affects human error, flow experience, and pleasant emotion, they can be controlled by adjusting the complexity of the task. Since target affects flow experience and work efficiency, target contributes to work efficiency. Target and complexity of the task are externally controllable factors. Therefore, organizations and supervisors can generate preferable flow experience by managing target and complexity of the task to improve work efficiency in order picking and reduce the human error of the worker.

### 4.3. Prediction of Pleasant Emotion, Arousal, and Engagement Using Deep Neural Networks

We clarify the prediction accuracy with deep neural networks which learned the time series features of the pulse wave sensor, the eye tracker, and the motion detectors when the worker evoked pleasant emotion, arousal, and engagement. In addition, we identify the important variables for predicting pleasant emotion, arousal, and engagement. We compared the accuracy of the generalized model with the accuracy of the personalized model. The comparison results show that we acquired knowledge for improving the accuracy of the generalized model.

There are two types of predictive models of supervised learning. The first type is the generalized model. The sample size was 544 (the number of subjects × the number of session = 17 × 32). The generalized model is constructed by training deep neural networks with the questionnaire results and time series features of workers other than the target person. The target person can receive prediction results simply by inputting the time series features to the generalized model. The second type is a personalized model. The personalized model is constructed by training deep neural networks with the questionnaire results and time series features of the target person by deep neural networks. Since the personalized model requires the target person to answer the questionnaire, the usability of the personalized model is less than that of the generalized model. However, the prediction accuracy of the personalized model is greater than that of the generalized model. Therefore, by analyzing the difference between the generalized and personalized models, we can acquire knowledge for improving the accuracy of the generalized model.

#### 4.3.1. Accuracy of Generalized Model

We constructed a generalized model by training deep neural networks with the questionnaire results and time series features of workers other than the target person. The target person can receive prediction results simply by inputting the time series features to the generalized model. The accuracy of the generalized model was verified. We conducted 17-fold-cross-validation with a worker as the test set and the other as the training set. The test set is the time series features of a subject. The training set is time series features of subjects other than the subject selected as the test set. The prediction accuracy is evaluated by the mean squared error normalized in Equation (2):*Error* = (*y*′ − *y*)^1/2^/(*ζ*(*y*′) − *η*(*y*′)),(2)
where *Error* is a value which divided a difference between the questionnaire result *y*′ and the predicted value *y* by a difference between the maximum value *ζ*(*y*′) and the minimum value of the questionnaire result *η*(*y*′). Mean squared error is traditionally used as an effective index for evaluating regression models. Therefore, in this research, the prediction model was evaluated using mean squared error.

In order to verify the effectiveness of variable selection in Section 3.2, all explanatory variables in Table 2 and Table 3 were input to deep neural networks, and the prediction results were shown in Figure 6. The accuracy of predicting pleasant emotion, arousal, and engagement by the deep neural networks is shown in Figure 7, Figure 8 and Figure 9. The prediction accuracy in Figure 7, Figure 8 and Figure 9 was improved by 0.05 maximum in pleasant, 0.05 in maximum in arousal, and up to 0.03 in engagement compared to the accuracy in Figure 6. Therefore, the variable selection in this research was effective for improving prediction accuracy. The prediction accuracy of pleasant emotion and arousal was highest when predicted from the time series features of the pulse wave sensor. The prediction accuracy of engagement was highest when predicted from the time series features of the motion detectors. The results were shown that the prediction accuracy among individuals was small when the pleasant emotion, arousal, and engagement were predicted by inputting time series features of motion detectors into generalized model.

We show important variables for predicting the pleasant emotion, arousal, and engagement in Table 5, Table 6 and Table 7. Table 5 shows the movements of the chest and waist are important variables for predicting the pleasant emotion. *f*(*e*(*a*^S3^), *x*) (*x* = 0.1, 0.15, 0.25, 0.3) in the pulse wave sensor at PL is smaller than that at UPL. Moreover, *σ*(*v*(*e*(*a*^S3^),0.25)) in pulse wave sensor at PL is greater than that at UPL. *f*(*r_z_*^BT^, *x*) (*x* = 7.5, 10, 12.5, 15, 17.5) in the motion detectors at PL is smaller than that at UPL. The rotation angle of the buttocks and the acceleration of the chest changed in conjunction with the movement of limb and arms during walking and order picking. *η*(*u*^PD^) in the eye tracker at PL is greater than that at UPL. Therefore, there is little unnecessary movement during order picking when a pleasant emotion is evoked. These time series features represent behavior control. The control of behavior is a factor that produces flow experience. It is suggested that deep neural networks predict the occurrence probability of the flow experience from the behavior control and indirectly predict pleasant emotions, arousal, and engagement as secondary reactions from the occurrence probability of the flow experience.

Table 2 and Table 3 list the abbreviations used in this study. The pulse wave sensor in Table 6 shows that *t*, *c*(*v*(*p*^RR^,0.5)), and *c*(*v*(*p*^HR^,0.25)) at arousal was smaller than that at sleepiness. *m*(*v*(*p*^HR^,0.5)) and *m*(*v*(*p*^HR^,0.9)) at arousal was greater than that at sleepiness. The eye tracker in Table 6 shows that *f*(*u_x_*^2D^, 6)), *f*(*u_x_*^2D^, 8)), and *m*(*v*(*a_z_*^HD^, 0.75)) at arousal was smaller than those at sleepiness. *m*(*v*(*a_z_*^HD^, 0.75)) and *m(u*^BK^) at arousal were greater than those at sleepiness. The motion detector in Table 6 shows that *m*(*r*^LA^) at arousal was greater than that at sleepiness. *c*(*v*(*r_z_*^LA^, 0.25)), *c*(*v*(*r_y_*^LA^, 0.75)), *c*(*v*(*r_z_*^LA^, 0.75)), and *η*(*r*^LA^) at the arousal were smaller than those of the sleepiness. The results show that the movement of the body at arousal was agiler than that at sleepiness. The agile movements have high exercise intensity and increase the heart rate of a worker. Therefore, the heart rate at the arousal temporarily increases. The peak-valley value is normalized to the maximum value. The temporary increase in heart rate caused the threshold of peak-valley to be high. As a result, many small peak-valley values were excluded. Therefore, the number of peak-valley detected was small.

The pulse wave sensor in Table 7 shows that psychological state of a worker was stable since *m*(*v*(*p*^LF^,0.25)), *m*(*v*(*p*^LF^,0.9)), *σ*(*v*(*p*^LF^,0.75)), *σ*(*v*(*p*^LF^,0.9)), and *σ*(*v*(*p*^HF^,0.9)) during concentration were smaller than those during boredom. The eye tracker in Table 7 shows that *f*(*u_x_*^2D^, 2)), *f*(*a_z_*^HD^, 35)), and *f*(*a_z_*^HD^, 30)) during concentration were greater than those during boredom. *f*(*u_x_*^2D^, 6)) and *f*(*a_y_*^HD^, 35)) during concentration were smaller than those during boredom. The motion detector in Table 7 shows that *f*(*r_z_*^BT^, 10) and *m*(*v*(*r_x_*^S2^, 0.25)) during concentration were greater than those during boredom. *m*(*v*(*r_y_*^SL^, 0.9)), *f*(*r_y_*^S1^, 5), *m*(*v*(*r_y_*^SL^, 0.25)), and *m*(*v*(*r_x_*^S2^, 0.25)) during concentration were smaller than those during boredom. These data show that there is little unnecessary movement in the order picking when the worker was engaged.

#### 4.3.2. Accuracy of the Personalized Model

We verified the prediction accuracy of the personalized model. We conducted 32-fold-cross-validation with an OP of the picker as the test set and the other as the training set. The test set is a time series features of a subject. The training set is time series features of the subjects other than the subject selected as the test set. The prediction accuracy is evaluated by the mean squared error normalized in Equation (2). Figure 10, Figure 11, Figure 12 and Figure 13 show the accuracy of the personalized model. The prediction accuracy of pleasant emotion and engagement was highest when predicted from the time series features of the motion detectors. The prediction accuracy of arousal was highest when predicted from the time series features of the eye tracker. The prediction accuracy of the personalized model was improved by 0.08 maximum in pleasant, 0.07 in maximum in arousal, and up to 0.05 in engagement compared to the generalized model. The results showed that the prediction accuracy among individuals was small when the pleasant emotion, arousal, and engagement were predicted by inputting time series features of motion detectors into personalized model. On the other hand, there were subjects with remarkably low accuracy of pleasant emotion when inputting the time series features of pulse wave sensor into the personalized model.

We show important variables for predicting pleasant emotion, arousal, and engagement using the personalized model in Table 8, Table 9 and Table 10. The tables show that both the generalized and personalized models selected the same variables as important. However, the accuracy of the personalized model was higher than that of the generalized model. The results showed that there are individual differences in the speed of walking and order picking.

## 5. Discussion

Deep neural networks have the power and ability of automatic feature extraction. However, from Figure 6, Figure 7, Figure 8 and Figure 9, automatic feature extraction does not work well, and the prediction accuracy with variable selection was better than the prediction accuracy without variable selection. The reason for this is that the amount of information included in the biological signal and behavior is less than the image showing the effectiveness in previous research [44]. Therefore, when inputting the time series features of biological signals and behavior to deep neural networks, it is necessary to select a variable.

In this research, we measured the level of pleasant emotion and arousal in 7-point scales and the level of engagement in 5-point scales. The spatial resolution of the questionnaire is 1. When the level of pleasant emotion and arousal differs by one-point, the mean squared error normalized is 0.14. When the level of engagement differs by one-point, the mean squared error normalized is 0.2. On the other hand, we showed that pleasant emotion, arousal, and engagement during order picking can be predicted from the behavior of the worker with an accuracy of error rate 0.12 or less. Considering these facts, this research can predict pleasant emotion, arousal, and engagement with high accuracy. In addition, the personalized model has sufficient accuracy to classify pleasant and unpleasant, arousal and sleepy, concentration and boredom.

From Figure 5, we predict pleasant emotion, arousal, and engagement as a secondary reaction of the flow experience. From Table 5, Table 6, Table 7, Table 8, Table 9 and Table 10 the worker in flow no longer experiences wasteful movements not required for work such as left arm and head. This fact shows that the flow state leads the worker to minimize the energy consumption of the body. As a result, worker enhances pleasure emotion, arousal, and engagement. Therefore, we predicted pleasant emotion, arousal, and engagement indirectly by time series features (power spectrum value and peak valley value) representing waste of behavior. On the other hand, personalized models are generally more accurate than generalized models. However, when time series features of the pulse wave sensor were input to deep neural networks, the accuracy of the personalized model was significantly lowered in the four subjects (C, F, K, and M) than in the generalized model. The reason for this is related to the characteristics of the pulse wave and the specificity of the subjects C, F, K and M. It is known that patterns of different emotions and biomedical signals appear in related studies even under the same stimulation depending on the situation. The related research pointed out that patterns of different vital signs appear depending on the situation even with the same stimulus. In addition, the time series features of the pulse wave appear to be complicated patterns due to the influence of exercise and the activity of the autonomic nervous system. Machine learning yields high accuracy with respect to objective variables and explanatory variable patterns present in the training data set, but accuracy for other patterns becomes unstable. In general, machine learning predicts highly accurately in patterns of target variables and explanatory variables present in the training set, but prediction accuracy in other patterns is unstable. Therefore, when the variation of emotion is large and the sample size is small, the number of patterns not existing in training data set increases. As a result, prediction accuracy decreases remarkably. Subjects C, F, K, and M have specificity in standard deviation of pleasant emotion, standard deviation of work efficiency, and average of flow state score (Figure 13). A small standard deviation of work efficiency shows that the worker has a high skill. C, F, and K have larger standard deviation of pleasant emotion compared to other subjects, and the skill of workers is high. M has larger standard deviation of pleasant emotion compared to other subjects, and the skill is low. In addition, since C, F, and M had a high average of flow state score, the standard deviation of pleasant emotion was caused by a flow experience. On the other hand, since the average of flow state score of K was remarkably low, the standard deviation of pleasant emotion was caused by factors other than the flow experience. The results in Figure 10 indicate that it is not possible to create a good predictive model for pleasant emotion by machine learning due to the small sample size when time series features of pulse waves of subjects with specificity were used. On the other hand, when pleasant emotion was predicted using the time series feature of behavior, the accuracy is stable and high. Therefore, it can be said that the sample size is sufficient when behavior is used as an index. From this, it is shown that if we take short emotions of work, we can predict emotion, arousal, and immersion stably by obtaining behavior from vital signs. In the case of using the time series feature of the pulse wave, as generalized model, the generalized model is more accurate, so let us learn data of similar users not only between individuals by multi task learning There is a high possibility that the accuracy can be improved. There was a difference between a person who is likely to increase the pulse by exercise and a person who is difficult to rise, and the accuracy became unstable. On the other hand, as Table 5 shows, there is a possibility that a remarkable decrease in precision due to characteristics of an individual hard to appear in a pulse wave can be solved by considering other person’s characteristics using multi task learning.

## 6. Conclusions and Future Work

The contributions of this research are as follows: First, we show that the organizations and supervisors can generate preferable flow experiences by giving a clear target and managing task complexity. In addition, pleasant emotions gave vitality to workers and decrease human error. Second, we showed that deep neural networks accurately predicted pleasant emotion, arousal, and engagement during order picking. The accuracy of the prediction model was the highest when predicting flow experience from the time series features of behavior.

In this research, the workers wore multiple devices. However, to put the monitoring system into practical use, it is necessary to decrease the number of devices because the multiple devices interrupt the order picking task and increase the cost of the monitoring system. One possible way to solve this dilemma is to install the sensors on the belt so that the worker can utilize the monitoring system without additional load.

The prediction accuracy of the generalized model was smaller than that of the personalized model. The prediction accuracy of the personalized model was the highest when predicting flow experience from the time series features of behavior. Since the important variables were the same for the generalized model and the personalized model, it was suggested that the picking speed, the walking speed, and the step are different for each individual. Therefore, by taking differences among individuals into consideration and conducting multi task learning, improvement of accuracy of the generalized model can be expected. In the future, we plan to devise measures to address these individual differences using multi-task learning. Since the psychological model constructed in this research is a short-term order picking, we will investigate the effect of long-term order picking.

## Figures and Tables

**Figure 1 sensors-19-00165-f001:**
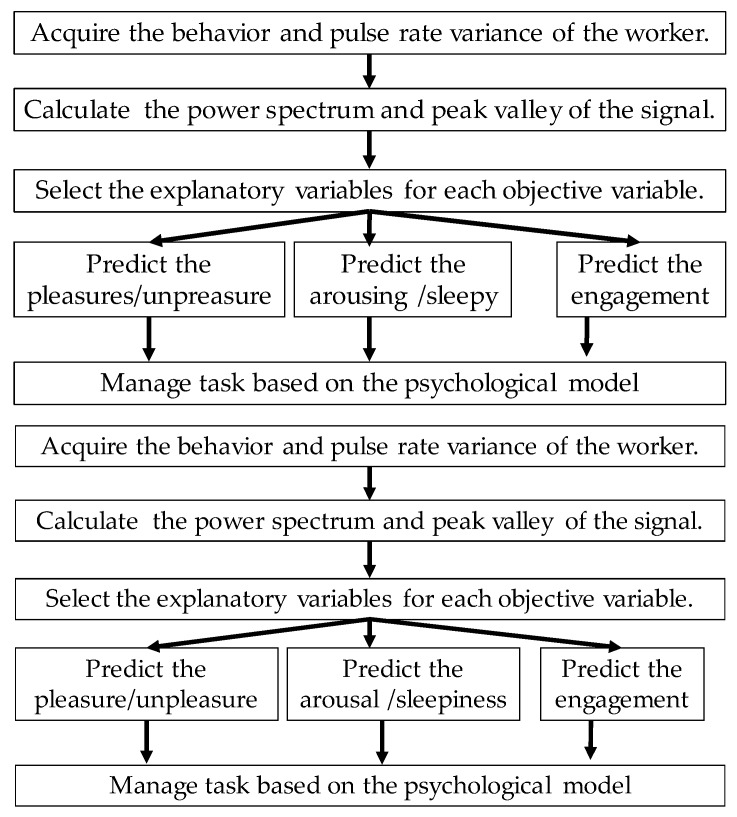
Task management using the prediction of pleasure, arousal, and engagement.

**Figure 2 sensors-19-00165-f002:**
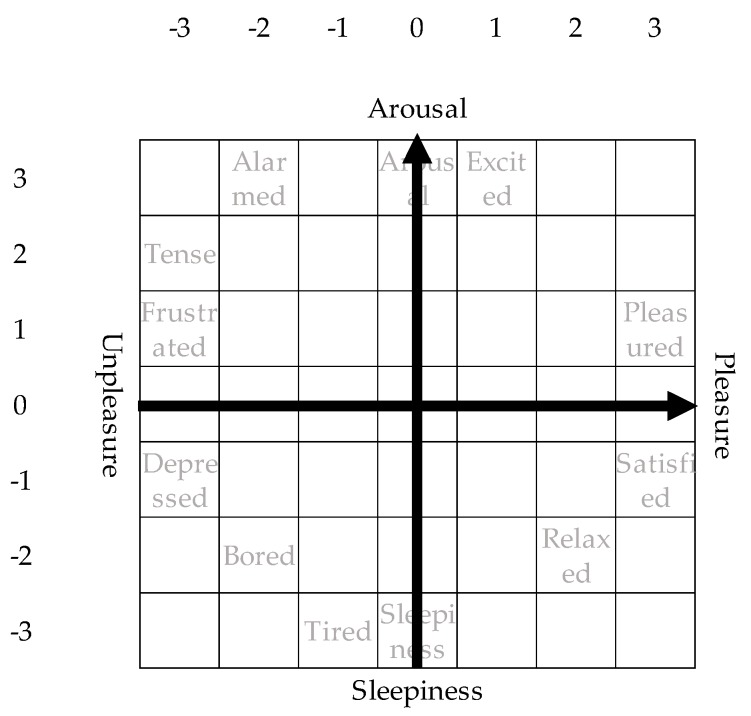
A questionnaire based on the circumplex model of affect.

**Figure 3 sensors-19-00165-f003:**
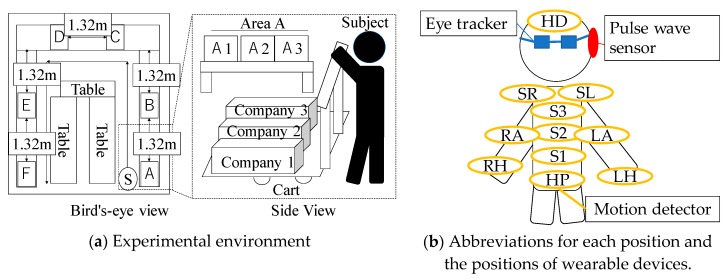
The experimental environment and the position of wearable devices.

**Figure 4 sensors-19-00165-f004:**
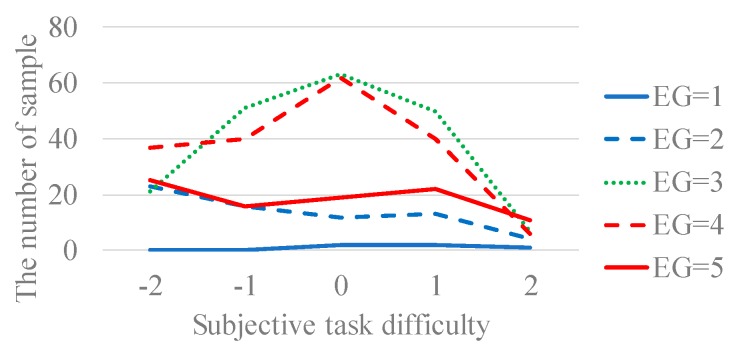
The sample size of engagement (EG) and subjective task difficulty.

**Figure 5 sensors-19-00165-f005:**
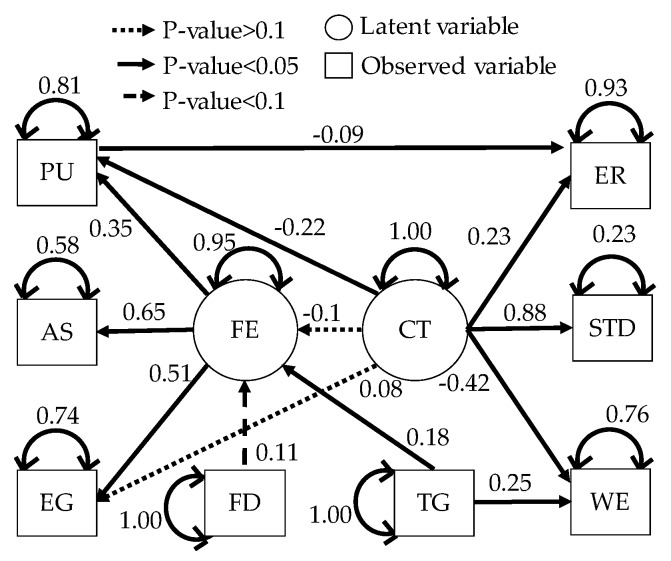
Covariance structure analysis result of the psychological model (PU: Pleasant emotion, AS: Arousal, EG: engagement, FE: flow experience, CT: complexity of the task, TG: target, FD: feedback, ER: human error, STD: subjective task difficulty, WE: work efficiency.).

**Figure 6 sensors-19-00165-f006:**
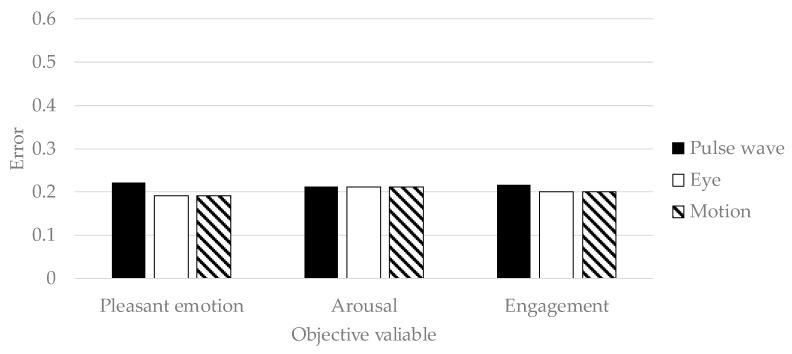
The accuracy of deep neural networks when inputted all time series features.

**Figure 7 sensors-19-00165-f007:**
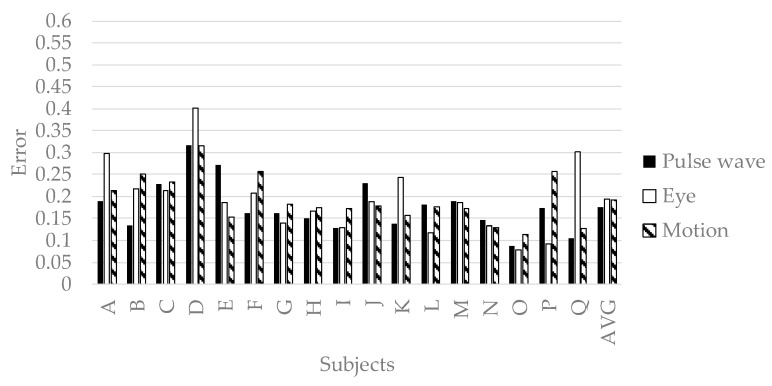
The accuracy of generalized model of pleasant emotion.

**Figure 8 sensors-19-00165-f008:**
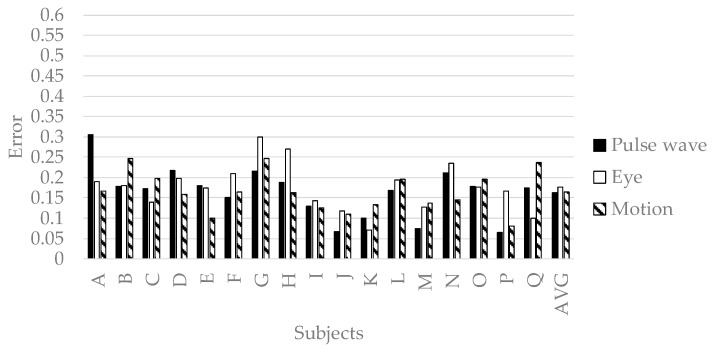
The accuracy of generalized prediction model of arousal.

**Figure 9 sensors-19-00165-f009:**
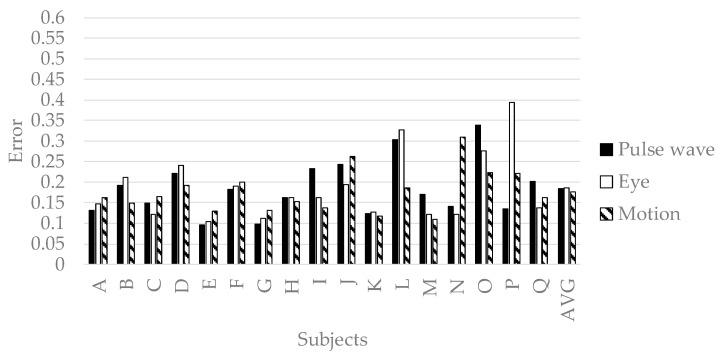
The accuracy of generalized model of engagement.

**Figure 10 sensors-19-00165-f010:**
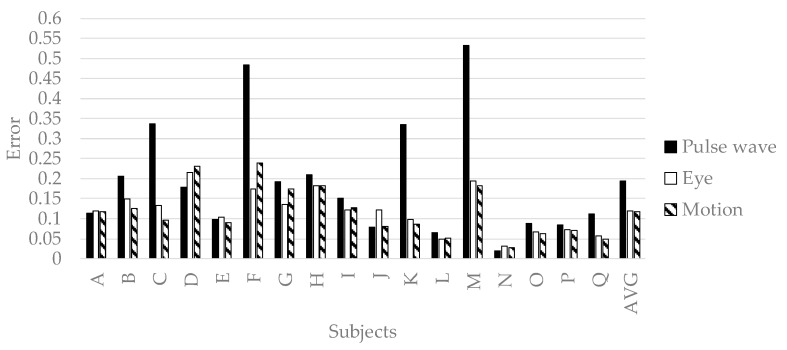
The accuracy of personalized model of pleasant emotion.

**Figure 11 sensors-19-00165-f011:**
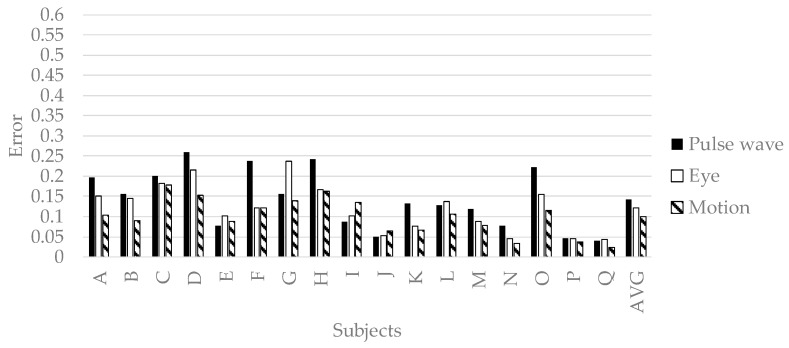
The accuracy of personalized prediction model of arousal.

**Figure 12 sensors-19-00165-f012:**
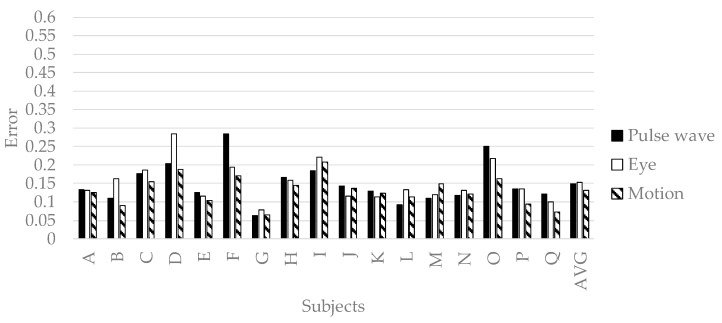
The accuracy of personalized model of engagement.

**Figure 13 sensors-19-00165-f013:**
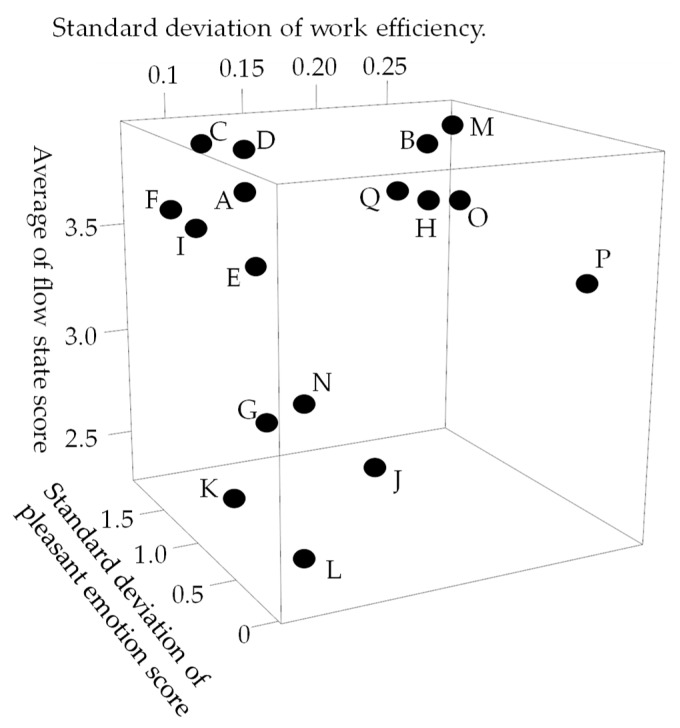
The accuracy of personalized model of pleasant emotion.

**Table 1 sensors-19-00165-t001:** Item of flow state scale.

Symbol	Item
*q* ^(1^ ^)^ _FE_	I felt that the task difficulty and worker skill of the work are balanced.
*q* ^(2)^ _FE_	I was fused with behavior and consciousness.
*q* ^(3)^ _FE_	I had a clear target.
*q* ^(4)^ _FE_	I had a clear feedback.
*q* ^(5)^ _FE_	I paid all the attention to the work.
*q* ^(6)^ _FE_	I had a sense of controlling the behavior.
*q* ^(7)^ _FE_	I could move my body naturally.
*q* ^(8)^ _FE_	I feel the transformation of time sense.
*q* ^(9)^ _FE_	I enjoyed a task.

**Table 2 sensors-19-00165-t002:** Time series features are calculated from time series data acquired with wearable devices.

Device	Types and Part of the Body		
	Power Spectrum Value	Peak Valley Value	Time
Pulse wave sensor	HF, LF, VLF, TP, Heart rate (HR), Lyapunov exponent, Entropy	VLF, LF, HF, TP, Heart rate (HR), R-R interval (RR), Lyapunov exponent, Entropy	-
Eye tracker	Eye movement in XYZ coordinate system(3D), Point of view(2D), Pupil diameter (PD), Triaxial accelerometer, Triaxial angular velocity	Eye movement in XYZ coordinate system(3D), Point of view(2D), Pupil diameter (PD), Triaxial accelerometer, Triaxial angular velocity	Eye movement, Eye blink (BK), Saccade
Motion detector	Left and right hands(LH, RH), Left and right elbow(LA, RA), Left and right shoulder(SL, SR), Buttocks(BT), Lumbar spine(S1), Thoracic spine(S2), Cervical spine(S3), head(HD) of the rotation angle	Left and right hands(LH, RH), Left and right elbow(LA, RA), Left and right shoulder(SL, SR), Buttocks(BT), Lumbar spine(S1), Thoracic spine(S2), Cervical spine(S3), head(HD) of the rotation angle	-

**Table 3 sensors-19-00165-t003:** Correspondence table of each feature quantity and mathematical symbol.

Mathematical Symbols	Meaning of Symbols
*m*(*data*)	Average of d time series data
*σ*(*data*)	Standard deviation of time series data
*η*(*data*)	Minimum value of time series data
ζ(*data*)	Maximum value of time series data
*c*(*data*)	Number of extremes of time series data
*v*(*data*, *threshold*)	Peak valley value of time series data when setting threshold
*f*(*data*, *frequency*)	Power spectrum value at frequency of time series data
*e*(*data*)	Entropy of time series data
*a* ^(*part*)^ *_axis_*	Acceleration of each axis of acceleration sensors attached to a part of the body
*g* ^(*part*)^ *_axis_*	Angular velocity of each axis of angular velocity sensors attached to a part of body
*r* ^(*part*)^ *_axis_*	Rotation angle of each axis of motion detectors attached to a part of body
*p* ^(*type*)^	Types of time series data acquired by the pulse wave sensor.
*u* ^(*type*)^	Types of time series data acquired by eye tracker

**Table 4 sensors-19-00165-t004:** The sample size of each square of circumplex of model of affect.

Arousal/Pleasure	−3	−2	−1	0	+1	+2	+3
+3	2	5	1	10	11	43	21
+2	4	4	28	31	36	79	7
+1	2	5	3	39	32	25	5
0	1	4	4	47	9	6	5
−1	2	3	15	15	9	3	4
−2	0	1	2	0	11	4	3
−3	0	0	0	2	0	0	0

**Table 5 sensors-19-00165-t005:** Top five of the important variables for predicting the pleasant emotion in the generalized model.

Pulse Wave Sensor			Eye Tracker			Motion Detector		
Variable	UPL	PL	Variable	UPL	PL	Variable	UPL	PL
*f*(*e*(*a*^S3^),0.1)	0.29	0.24	*η*(*u*^PD^)	2.91	3.12	*f*(*r_z_*^BT^, 17.5)	2.46	0.77
*f*(*e*(*a*^S3^),0.15)	0.16	0.12	*m*(*v*(*u_z_*^3D^, 0.9))	457	415	*f*(*r_z_*^BT^, 15)	2.85	0.87
*f*(*e*(*a*^S3^),0.25)	0.09	0.06	*m*(*u*^PD^)	3.79	3.98	*f*(*r_z_*^BT^, 12.5)	3.31	1.07
*f*(*e*(*a*^S3^),0.3)	0.07	0.05	*σ*(*v*(*u_z_*^3D^, 0.75))	417	435	*f*(*r_z_*^BT^, 10)	4.23	1.42
*σ*(*v*(*e*(*a*^S3^),0.25))	4.42	3.17	*m*(*v*(*u_z_*^3D^, 0.75))	523	461	*f*(*r_z_*^BT^, 7.5)	5.94	2.06

**Table 6 sensors-19-00165-t006:** Top five of the important variables for predicting the arousal in the generalized model.

Pulse Wave Sensor			Eye Tracker			Motion Detector		
Variable	SE	AR	Variable	SE	AR	Variable	SE	AR
*t*	47.3	43	*f*(*u_x_*^2D^, 6))	1039	948	*c*(*v*(*r_z_*^LA^, 0.75))	82.1	55.9
*c*(*v*(*p*^RR^,0.5))	5	3.64	*f*(*u_x_*^2D^, 8))	6238	5162	*c*(*v*(*r_y_*^LA^, 0.75))	69.5	51.7
*m*(*v*(*p*^HR^,0.9))	902	966	*m*(*v*(*a_z_*^HD^, 0.75))	2.84	3.9	*m*(*r*^LA^)	−39.1	−20.4
*m*(*v*(*p*^HR^,0.5))	892	945	*m(u*^BK^)	326	182	*c*(*v*(*r_z_*^LA^, 0.25))	220	194
*c*(*v*(*p*^HR^,0.25))	13.2	11.1	*f*(*a_z_*^HD^, 45))	455	235	*η*(*r*^LA^)	−81.1	−82.1

**Table 7 sensors-19-00165-t007:** Top five of the important variables for predicting the engagement in the generalized model.

Pulse Wave Sensor			Eye Tracker			Motion Detector		
Variable	BR	CR	Variable	BR	CR	Variable	BR	CR
*m*(*v*(*p*^LF^,0.9))	2325	1484	*f*(*u_x_*^2D^, 2))	942	1664	*f*(*r_z_*^BT^, 10)	1.4	1.81
*σ*(*v*(*p*^LF^,0.9))	3695	2711	*f*(*u_x_*^2D^, 6))	1068	944	*f*(*r_y_*^BT^, 5)	2.4	2.35
*m*(*v*(*p*^LF^,0.25))	2620	1708	*f*(*a_z_*^HD^, 35))	333	440	*m*(*v*(*r_y_*^SL^, 0.9))	0.49	0.2
*σ*(*v*(*p*^HF^,0.9))	4782	3317	*f*(*a_z_*^HD^, 30))	492	664	*m*(*v*(*r_y_*^SL^, 0.25))	0.36	0.14
*σ*(*v*(*p*^LF^,0.75))	3570	2633	*f*(*a_y_*^HD^, 35))	238	223	*m*(*v*(*r_x_*^S2^, 0.25))	1.57	3.39

**Table 8 sensors-19-00165-t008:** Top five of the important variables for predicting the pleasant emotion in the personalized model.

Pulse Wave Sensor			Eye Tracker			Motion Detector		
Variable	UPL	PL	Variable	UPL	PL	Variable	UPL	PL
*σ*(*v*(*e*(*a*^S3^),0.9))	0.39	0.30	*η*(*u*^PD^)	2.91	3.12	*f*(*r_z_*^BT^, 15)	2.85	0.87
*σ*(*v*(*e*(*a*^S3^),0.75))	0.36	0.28	*m*(*u*^PD^)	3.79	3.98	*f*(*r_z_*^BT^, 17.5)	2.46	0.77
*f*(*e*(*a*^S3^),0.25)	0.07	0.05	*m*(*v*(*u_z_*^3D^, 0.75))	523	461	*f*(*r_z_*^BT^, 10)	4.23	1.42
*f*(*e*(*a*^S3^),0.15)	0.16	0.12	*m*(*v*(*u_z_*^3D^, 0.9))	457	415	*f*(*r_z_*^BT^, 7.5)	5.94	2.06
*m*(*v*(*e*(*a*^S3^),0.9))	4.85	3.55	ζ(*u_z_*^3D^)	851	769	*f*(*r_z_*^BT^, 5.0)	10.5	3.82

**Table 9 sensors-19-00165-t009:** Top five of the important variables for predicting the arousal in the personalized model.

Pulse Wave Sensor			Eye Tracker			Motion Detector		
Variable	SE	AR	Variable	SE	AR	Variable	SE	AR
*c*(*v*(*p*^RR^,0.5))	5	3.64	*f*(*g_x_*^HD^, 15))	1040	949	*c*(*v*(*r_y_*^LA^, 0.75))	69.5	51.7
*t*	47.3	43	*T*	47.3	43	*c*(*v*(*r_y_*^LA^, 0.5))	140	113
*m*(*v*(*p*^HR^,0.5))	892	945	*u* ^BK^	327	182	*c*(*v*(*r_x_*^LA^, 0.5))	123	101
*m*(*v*(*p*^HR^,0.9))	902	966	*f*(*a_z_*^HD^, 50))	406	208	*c*(*v*(*r_x_*^LA^, 0.25))	204	180
σ(*p*^HR^)	410	512	*f*(*a_z_*^HD^, 35))	683	384	*m*(*v*(*r_x_*^LA^, 0.9))	−38.4	−28.2

**Table 10 sensors-19-00165-t010:** Top five of the important variables for predicting the engagement in the personalized model.

Pulse Wave Sensor			Eye Tracker			Motion Detector		
Variable	BR	CR	Variable	BR	CR	Variable	BR	CR
*σ*(*v*(*p*^LF^,0.9))	3695	2711	σ(*v*(*a_y_*^HD^, 0.25))	2.39	2.08	*m*(*v*(*r_y_*^SL^, 0.9))	0.49	0.2
*m*(*v*(*p*^LF^,0.9))	2325	1484	*m*(*a_z_*^HD^)	5.80	5.27	*σ*(*v*(*r_x_*^SL^, 0.25))	4.34	3.87
*σ*(*v*(*p*^LF^,0.75))	3570	2633	σ(*v*(*a_y_*^HD^, 0.5))	1.94	1.73	*m*(*v*(*r_y_*^SL^, 0.25))	0.36	0.14
*m*(*v*(*p*^LF^,0.25))	2620	1708	*m*(*v*(*a_y_*^HD^, 0.5))	−6.58	−7.08	σ(*r*^S3^)	0.29	0.09
*m*(*v*(*p*^LF^,0.75))	2315	1454	*m*(*v*(*a_y_*^HD^, 0.25))	−7.26	−7.66	*m*(*v*(*r_x_*^S2^, 0.25))	1.57	3.39

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
