# Peer review of "Predicting Emotion and Engagement of Workers in Order Picking Based on Behavior and Pulse Waves Acquired by Wearable Devices"

_sensors, 2019, doi:10.3390/s19010165_

Reviewer 1 Report

Topic is definitely interesting and has practical value for any kind of industry requiring human labor. There is a lot of irrelevant references (motion capture with Internet behavior, irrelevant HUD usecases) that are too far from the particular research and should be considered to be removed. Text is somehow difficult to follow, this might be because of language issues, insufficient explanation of tables' content and relevance. Fig 6 is perhaps the most important/conclusive figure of the paper and should somehow present the distribution of classification results (3D distribution graph of individual measurements?). Conclusions should contain more (conclusive) information about the outcomes, perhaps proposals how to reduce the error rate 20%, exclude new references. Authors personal opinion is that number of trial users of 17 individuals might be too small for a such complex task as emotional condition assessment and, perhaps, more professional psychological condition assessment is needed for the reference instead of simple questionnaire.

Author Response

Thank you very much for your review and helpful advice. We corrected the paper considering the points that were pointed out. 

Reviewer 2 Report

My major concern is the presentation of this paper. To me, it is far from ready to be published. There so many grammar mistakes that make the reading difficult. For instance:

The caption of Table 3: "The sample size of each square of circumplex of model of affect". This is confusing.

Again: "In this study, to clarify the mechanism of occurrence of emotion and engagement during order picking." This is not a complete sentence.

Why Table 4 is separated into two different pages?

In many places, the authors just recap/describe the values listed in the Figure/Table without any in-depth analysis. For instance, the entire paragraph from line 356 to line 367. Those information can be easily captured from the Figure/Table if necessary.

2. The discussion of related works is insufficient. Please also summary the works that using body wearable sensors for emotion detection.

3. It would be better to give more details regarding the deep neural network model.

Author Response

(The authors gave the same response as above.)

Reviewer 3 Report

The article was about predicting emotion and engagement of worker in order picking based on wearable sensors. Therefore, the literature review should include articles where wearable sensors are used to detect emotions, in the current form the literature review is not sufficient. See For instance:
https://www.researchgate.net/publication/329116315_Wearable_affect_and_stress_recognition_A_review

Assessing real-time cognitive load based on psycho-physiological measures for younger and older adults

E Ferreira, D Ferreira, SJ Kim, P Siirtola, J Röning, JF Forlizzi, AK Dey

In page 3, line 111: What is PANAS?

page 4: line 120 & line 122: repetition

In chapter 3.1 you should introduce the used devices, brand, what they can measure, sampling frequency, ...

Deep Learning is used in the article, and features given as inputs to the deep learning algorihms are listed in Table 2. One of the main advantages of using deep learning is that it can select features automatically, and therefore, the person training the models do not need to select features. Why in this case features were given as inputs to DL algorithm and not the raw data?

page 6, line 210: did persons have any previous experience in order picking?

Figure 6: only mean accuracies are given, would like to know the variance of accuracy between 17 study subjects

Table 5 is not cited in text. What is for instance region s3?

Table 4 & 5 are too difficult to read, almost impossible. Please, modify them to make them more readable.

Extensive editing of English language and style required

Author Response

Thank you very much for your review and helpful advice. We corrected the paper considering the points that were pointed out.

Round  2

Reviewer 1 Report

Paper has been improved in all aspects requested. Error in reference on line 155. No further suggestions.

Author Response

(The authors gave the same response as above.)

Reviewer 2 Report

1. There are many abbreviations used without explanation. For instance, in table 2, HF, LF, VLF, TP appear at line 146. But the explanations come very late at line 178.

2. Some of the descriptions are confusing, for instance:

In line 162: 

"The sampling rate of the pulse wave sensor was 1000 Hz. The sampling rate of the VLF, LF, HF, TF, Lyapunov exponent and entropy was 1 Hz. The motion detector is Neuron made by NOITOM. The sampling rate of the motion detector was 50 Hz."

In table 2, it seems that HF and TF are the features extracted from the time series of pulse wave sensor. Then, what do you mean by "The sampling rate of the pulse wave sensor was 1000 Hz. The sampling rate of the VLF, LF, HF, TF, Lyapunov exponent and entropy was 1 Hz"?

Also, what do you mean by 'Types and Region' in Table 2?

3. Follow comments 3:

As you have extracted time series features, what are the sliding window length you have used? are overlapping between the adjacent windows?

In addition, since different sampling frequencies were used for the three different sensors, what will be the impact of sampling frequency on the recognition performance?

4. One of the previous comment was "given the power and ability of deep learning in auto feature extraction, what are the benefits of the feature engineering study in this paper?" what will be the accuracy without the use of the features listed in Table 2 and 3?

5. What was the motivation of using 'Error' (mean squared error) as the performance metric? Can you explain? It would be better to provide the results of True-positive rate, False-negative, and F1-score.

6. Based on the results exhibit in Figure 6-8, it seems that the accuracy (i.e., True positive rate) ranges from 70%~80%, this is not very promising. Any explanation regarding this? 

7. It would be better to combine/move the content in the appendix section with the evaluation. Or explain at the beginning of the appendix about the motivation.

Author Response

(The authors gave the same response as above.)

Reviewer 3 Report

Line 155: citation broken

Line 393: σ(v(e(aS3),0.25)) this kind of equations are impossible to read. According to Table 3, σ states for standard deviation, but where can I find v, e, a, and S3? You really need to rethink these notions. Reader cannot find explanation to these. Open these, and tell the reader where to find explanations to all of these variables.

Author Response

Thank you very much for your review and helpful advice. We corrected the paper considering the points that were pointed out. 

Round  3

Reviewer 2 Report

I appreciate the authors' effort in revising the draft. I have no further comment.